# Smoking trajectories and risk of stroke until age of 50 years – The Northern Finland Birth Cohort 1966

**Ina Rissanen** [1,2,3,4]*, **Petteri Oura** [3,5], **Markus Paananen** [3,5], **Jouko Miettunen** [3,5], **Mirjam I. Geerlings** [4]

**1** Department of Neurology, Oulu University Hospital, Oulu, Finland, **2** Department of Neurosurgery, Oulu University Hospital, Oulu, Finland, **3** Medical Research Center Oulu, Oulu University Hospital and University of Oulu, Oulu, Finland, **4** Julius Center for Health Sciences and Primary Care, University Medical Center Utrecht and Utrecht University, Utrecht, The Netherlands, **5** The Center For Life Course Health Research, Faculty of Medicine, University of Oulu, Oulu, Finland

* ina.rissanen@oulu.fi

**Data Availability Statement:** Data is available from the Northern Finland Birth Cohort (NFBC) for researchers who meet the criteria for accessing confidential data. Please, contact NFBC project

## Abstract

### Background

Smoking is a well-known risk factor for stroke. However, the relationship between smoking trajectories during the life course and stroke is not known.

### Aims

We aimed to study the association of smoking trajectories and smoked pack-years with risk of ischemic and haemorrhagic strokes in a population-based birth cohort followed up to 50 years of age.

### Methods

Within the Northern Finland Birth Cohort 1966, 11,999 persons were followed from antenatal period to age 50 years. The smoking behaviour was assessed with postal questionnaires at ages 14, 31 and 46 years. Stroke diagnoses were collected from nationwide registers using unique study number linkage. The associations between smoking behaviour and stroke risk were estimated using Cox regression models.

### Results

Six different patterns in smoking habits throughout the life course were found in trajectory modelling. During 542,140 person-years of follow-up, 352 (2.9%) persons had a stroke. Continuous smoking during the life course was associated with increased stroke risk (HR = 1.69; 95% CI 1.10–2.60) after adjusting for sex, educational level, family history of strokes, leisure-time physical activity, body mass index, alcohol consumption, hypertension, hyper-cholesterolemia, and diabetes. Per every smoked pack-year the stroke risk increased 1.04-fold (95% CI 1.03–1.06). Other smoking trajectories were not significantly associated with stroke risk, nor were starting or ending age of smoking.

center (NFBCprojectcenter@oulu.fi) and visit the cohort website (www.oulu.fi/nfbc) for more information.

**Funding:** NFBC1966 received financial support from University of Oulu [grant no. 65354, 24000692]; Oulu University Hospital [grant no. 2/97, 8/97, 24301140]; Ministry of Health and Social Affairs [grant no. 23/251/97, 160/97, 190/97]; National Institute for Health and Welfare, Helsinki [grant no. 54121]; Regional Institute of Occupational Health, Oulu, Finland [grant no. 50621, 54231]; and ERDF European Regional Development Fund [grant no. 539/2010 A31592]. This work was supported by The Research Foundation of the Pulmonary Diseases (Hengityssairauksien tutkimussäätiö), Finland, The Maire Taponen Foundation (Maire Taposen säätiö), Finland, and The Jalmari and Rauha Ahokas Foundation (Jalmari ja Rauha Ahokkaan säätiö), Finland. The funders had no role in study design, data collection and analysis, decision to publish, or preparation of the manuscript.

**Competing interests:** The authors have declared that no competing interests exist.

**Abbreviations:** BMI, body mass index; ICD, International Classification of Disease; ICH, intracerebral haemorrhage; IS, ischemic stroke; NFBC, Northern Finland birth cohort; SAH, subarachnoid haemorrhage; TIA, transient ischemic attack.

## Conclusion

Accumulation of smoking history is associated with increased risk of stroke until age of 50 years. The increased stroke risk does not depend on the age at which smoking started. Given that the majority starts smoking at young age, primary prevention of strokes should focus on adolescent smoking.

## Introduction

Stroke is the second largest cause of death and third largest cause of disability, accounting for 10% of the disease burden in the world[1]. While the overall incidence of stroke is declining in developed countries, the incidence of stroke among people under 50 years of age, is increasing [1–4]. Of all strokes, about 80% are ischemic and 20% haemorrhagic. However, the proportion of haemorrhagic strokes, i.e. intracerebral haemorrhage (ICH) and subarachnoid haemorrhage (SAH), is higher among people under 50 years of age[5] About a quarter of ischaemic strokes and half of haemorrhagic strokes occur in people younger than 65 years of age[6]. Young stroke patients usually have better outcome after stroke than older patients[7,8]. Nonetheless, the risk for recurrent strokes and other cardiovascular events remain high among young stroke patients even for decades after the first stroke[9,10].

Smoking is a well-known risk factor for stroke both under and over 50 years of age[11,12], causing about one fifth of the total stroke burden[13]. The stroke risk is especially associated with current smoking, with a clear decline in stroke risk among individuals who quit smoking [14,15]. Previous studies have also shown a dose-dependent association between smoking and risk of stroke[16,17]. However, it is not known to what extent the association is different depending on the age at which smoking started or ended. Previous studies concerning adolescence smoking and stroke risk have assumed the smoking behavior to remain stable during the life course[18,19]. Nonetheless, it has been shown that after adolescents initiate smoking, they have diverse smoking trajectories[20–23].

Previous studies on smoking and risk of stroke have been collected retrospectively or had a follow-up period less than 50 years, and therefore have not been able to study the association between smoking trajectories and stroke risk throughout the life course[24–27]. The risk estimates for stroke may be dependent on the length of follow-up and age when smoking was measured. In a previous study of different risk factors of stroke, the effect of smoking on stroke risk depended on the follow-up time[28]. Previous studies suggest that the effect of smoking on stroke risk seems to be greater at younger age[29]. However, previous studies have included only adult subjects, and therefore, have not been able to study the effect of smoking on stroke risk starting from adolescence. For stroke prevention, it is important to investigate the life course smoking trajectories and stroke risk especially at young age.

## Aims

We aimed to examine the relationship between 1) smoking trajectories, 2) smoked pack-years, and 3) starting and ending age of smoking, and the risk of ischemic or non-traumatic haemorrhagic stroke in a prospective population-based birth cohort study in Finland. A group-based trajectory modelling technique was used to investigate the smoking trajectories from 5 to 46 years of age, capturing the entire developmental course of smoking on the stroke risk at young age.

## Methods

### Design and population of the study

The study population is the Northern Finland Birth Cohort 1966 (NFBC1966), an unselected population-based birth cohort containing data on 12,058 babies born alive in the Finnish provinces of Oulu and Lapland with an expected date of birth in 1966. Permission to gather data was obtained from the Ministry of Social Affairs and Health, and the study was approved by the Ethical Committee of Northern Ostrobothnia Hospital District in Oulu, Finland. Data protection was scrutinized by the Privacy Protection Agency of Finland. Informed consent was inquired from all the participants. Subjects who declined use of their data (n = 59) were excluded from the study. Reasons why participants declined use of their data were unknown. The sample for the current study included 11,999 subjects who were followed from birth date to their first stroke, death, moving permanently abroad, or to 31st of December 2015. The mean follow-up time per participant was 45.2 years (standard deviation (SD) 12.0 years), and the median follow-up time 49.4 years. The 17 persons who moved abroad and whose moving date was not known were assumed to be followed until end of 2015.

Data collection of NFBC1966 started in the year 1965 when the mothers were pregnant. Data on the individuals born into this cohort, and on their parents, were collected since the 24th gestational week. At the age of 14 years the first follow-up was conducted by a postal questionnaire concerning cohort members' growth, health, living habits, school performance and family situation (n = 11,010; 93.6%). At the age of 31 years a postal questionnaire (n = 8,767; 77.4%) and a clinical health examination (n = 6,033; 71.3%) were conducted for cohort members to study their health, physical performance capacity, occupation and working history, use of public health services, and living habits. The same factors were assessed also at age 46 years with a postal questionnaire (n = 6,868; 66.5%) and a clinical examination (n = 5,861; 56.7%). Reasons for non-participation later in the study were unknown. Existing nationwide registers, *i.e.* Care Register for Health Care, Causes of Death Register, and register of medication reimbursement, were linked to questionnaire and clinical examination data with personal identification numbers. The linkage was fully complete for stroke diagnoses.

### Smoking

Smoking was the studied exposure in this study. Smoking habits of subjects were asked in questionnaires at ages 14, 31, and 46 years. At age 14 the smoking habit was asked with options *'I have never tried'*, *'I tried once'*, *'I have tried twice or more'*, *'I smoke occasionally'*, *'I smoke about twice a week'*, *'I smoke 1–5 cigarettes daily'*, *'I smoke 6–10 cigarettes daily'*, and *'I smoke more than 10 cigarettes daily'*. Persons smoking twice per week or more were categorized to be smoking at that age. At age 31 and 46, smoking status assessment was based on standards of the Finland Cardiovascular Risk (FINRISK) study and its four smoking-related questions ('*Have you ever smoked*?', '*Have you ever smoked regularly, almost daily for at least a year*?', '*Do you smoke now*?' and '*When was the last time you smoked*?'). Those who answered 'yes' to a question 'Do you smoke now?' were considered to be smoking at that age. The starting and ending ages of smoking were asked both at age 31 and 46. The 31 years follow-up assessment was selected as the primary source for data on smoking starting age and the 46 years follow-up assessment was used as the primary source for data on smoking ending age to reduce recall bias. The binary smoking status of each year in persons' life, except for ages 14, 31, and 46, was calculated from the smoking starting and ending ages. This information was used in the trajectory model together with binary smoking status for ages 14, 31, and 46 which were collected from questionnaires.

Smoked pack-years by age 31, between ages 31 and 46, and by age 46 were calculated from questionnaires at ages 31 and 46. Calculation of pack-years was based on information on smoking starting and ending ages and the question *'How much per day do you usually smoke now or smoked before you gave up smoking?'*. Number of years each subject had smoked were calculated by subtracting starting age from ending age.

## Strokes

Stroke was the outcome variable of this study. Strokes were identified from national Care Register for Health Care or Causes of Death Register and classified by primary diagnosis (Table 1). Ischemic stroke and transient ischemic attack (TIA) were considered as 'ischemic strokes' and SAH and ICH as 'haemorrhagic strokes'. For analyses of any stroke type, ischemic strokes, haemorrhagic strokes, and other cerebrovascular diseases were combined. Traumatic SAH, traumatic ICH, epidural hematoma or subdural hematoma were not included in the series. The diagnostic coding has been based on the WHO international classification of diseases (ICD) in Finland since 1967 [30]. Subjects having two or more stroke diagnoses were classified by primary diagnosis.

Subjects were also asked in the 46-years follow-up questionnaire if they had a stroke diagnosed by a physician. Self-reported strokes were classified as other cerebrovascular diseases if the subject did not have diagnoses in registers (n = 13). The age of stroke onset was not known for the subjects with self-reported stroke who were not present in registers.

## Covariates

Information on covariates was collected from follow-up questionnaires, clinical examinations, and national registers at age of 46 years. Educational level was classified into basic ($\leq$ 9 years; comprehensive school), secondary (9–12 years; upper secondary school or vocational school) and tertiary ($>$ 12 years; university or university of applied sciences) education by the highest self-reported education achieved in the questionnaire. The family history of strokes among 1st degree relatives was also self-reported in the questionnaire. Leisure-time physical activity was measured as how many hours exercise the subject had per month according to questions *'How often do you exercise on your leisure-time a) with low intensity and b) with high intensity'* and *'What is the duration of each exercise a) with low intensity and b) with high intensity'*. Both low and high intensity exercises were considered equally. Weight and height were measured during the clinical examination and asked with the postal questionnaire. BMI ($kg/m^2$) was calculated from weight and height using the measured values as primary source. Daily mean alcohol consumption (g/day) was calculated from self-reported questionnaire data.

**Table 1. Classification of strokes according to International Classification of Diseases 8, 9 and 10.**

|  | ICD-8 codes | ICD-9 codes | ICD-10 codes |
|---|---|---|---|
| SAH | 430 | 430 | I60, I69.0 |
| ICH | 431 | 431 | I61, I69.1 |
| IS | 432–434 | 433–434 | I63, I69.3 |
| TIA | 435 | 435 | G45 |
| Other cerebrovascular diseases | 436–438 | 436–438 | I64-I68, I69.4, I69.8, G46 |
| Years used in Finland | 1967–1986 | 1987–1995 | 1996- |

ICD = International Classification of Diseases, SAH = subarachnoid haemorrhage, ICH = intracerebral haemorrhage, IS = ischemic stroke, TIA = transient ischemic attack. Stroke syndromes (ICD-9 code 438; ICD-10 code G46) were classified according to etiological sub-code (ICD-9 codes 430–437; ICD-10 codes I60-I67) or as other cerebrovascular diseases if sub-codes were not present.

Hypertension was defined as having a mean of measured systolic blood pressure $\geq 140$ mmHg, diastolic blood pressure $\geq 90$ mmHg, self-reported diagnosis of hypertension, or using antihypertensive medication (ATC codes C02 antihypertensives, C03 diuretics, C07 beta blocking agents, C08 calcium channel blockers, and C09 agents acting on the renin-angiotensin system) according to national register of medication reimbursement. The presence of hypercholesterolemia was noted in case of triglyceride level $> 2.0$ mmol/l, LDL-cholesterol $> 3.0$ mmol/l, HDL cholesterol $< 1.0$ mmol/l, or in case of current lipid-lowering therapy (ATC code C10 lipid modifying agents) in register. Diabetes mellitus was diagnosed in the presence of fasting blood glucose level $\geq 7.0$ mmol/l, blood glucose level of $\geq 11.1$ mmol/l after two hours 75g oral glucose tolerance test, $HbA_{1C} \geq 48$ mmol/mol (6.5%), self-reported type 1 or type 2 diabetes, or by the use of antidiabetic therapy (ATC code A10 drugs used in diabetes) in register.

## Statistical analyses

Multiple imputation was used to impute missing data for independent variables of planned analyses. It included independent variables (smoking status and smoked pack-years), covariates, outcome variables and other variables used only as predictors for multiple imputation. The outcome variables for stroke diagnoses were complete for all 11,999 subjects and were not imputed. All 35 variables included in the multiple imputation procedure and rates of missing data for each variable are listed in S1 Table. Data were missing both due to loss of follow-up of subjects and due to missing measurements of available subjects. The overall amount of incomplete data was 27.5%, and therefore, multiple imputation was conducted 30 times. Data were assumed to be missing at random. Model for scale variables was linear regression and for nominal variables logistic regression. The pooled results were reported in the analyses. IBM SPSS Statistics 24 were used for multiple imputation.

To reveal latent trajectories in the smoking data, SAS version 9.4 (SAS Institute Inc., Cary, NC, USA) and the PROC TRAJ latent class growth modelling (LCGM) macro[31,32] were used. LCGM is semi-parametric modelling approach which aims to detect classes of individuals which share a similar pattern of change (*i.e.* trajectory) over time[33]. Information on smoking status for each age between 5 and 46 years were used in trajectory modelling. Due to binary data (smoking vs. non-smoking), we used the logit-based (LOGIT) model in the PROC TRAJ. Models with one to seven trajectory classes were tested, and the selection of six trajectory classes as the most suitable model was based on the following measures of model adequacy which are shown for each tested model in Results Table 2: 1) Bayesian Information Criterion (BIC) and Akaike Information Criterion (AIC), where lower absolute values indicate better fit of data; 2) the Bayes Factor ($B_{10}$) and the log form of the Bayes Factor ($2\log_e(B_{10}) \approx 2(\Delta BIC)$), where $\Delta BIC$ is the BIC of the alternative (*i.e.* more complex) model less the BIC of the null (*i.e.* less complex) model; $2\log_e(B_{10})$ is interpreted as the degree of evidence favoring the alternative model ($> 6$ indicates strong evidence against the null model); 3) posterior membership probabilities, where class averages of $> 0.70$ are considered acceptable; and 4) absolute and relative class sizes, also taking into consideration the subsequent analyses[31–34]. After the best model was selected, each subject was assigned to the trajectory class with the highest posterior membership probability[32,33]. The reference group with lowest smoking prevalence was named as 'non-smokers', even though some individuals in that smoking trajectory class had smoked during their life.

Difference in mean age of onset between ischemic and haemorrhagic strokes was studied with independent samples t-test. The incidences of strokes according to smoking trajectory classes were calculated. The associations (hazard ratios (HR) with 95% confidence intervals

**Table 2. Model fit parameters of trajectory models with 1–6 classes.**

| Number of Classes | AIC | BIC | Null Model | $2\log_e(B_{10})$ | Sample size (%) | Average posterior probability |
|---|---|---|---|---|---|---|
| 1 | -8285640.11 | -8285661.96 | - | - | 100 | 1.00 |
| 2 | -5200634.25 | -5200685.23 | 1 | 6169953.46 | 73.6/26.4 | 1.00/0.99 |
| 3 | -3785208.66 | -3785288.76 | 2 | 2830792.94 | 36.3/40.1/23.6 | 0.99/0.99/1.00 |
| 4 | -3487005.04 | -3487114.27 | 3 | 596348.98 | 34.8/35.5/11.4/18.3 | 0.99/0.99/0.97/1.00 |
| 5 | -3276003.76 | -3275865.40 | 4 | 422497.74 | 34.7/35.8/9.7/15.7/4.1 | 0.98/0.99/0.98/0.99/0.96 |
| 6 | -3105916.25 | -3106083.74 | 5 | 339563.32 | 27.6/27.8/9.2/16.6/10.8/8.0 | 0.97/0.99/0.98/0.97/0.98/0.95 |

AIC = Akaike information criterion, BIC = Bayesian information criterion, $B_{10}$ = Bayes factor, $2\log_e(B_{10})$ = The log form of the Bayes factor; interpreted as the degree of evidence favoring the alternative model ($> 6$ indicates strong evidence against the null model)[31], n.a. = not applicable. Sample size showed per class based on most likely class membership. Average posterior probability for individuals belonging to the respective class. The seven-class model failed to converge.

(95% CI)) between smoking trajectory classes, smoked pack-years, starting age of smoking and ending age of smoking with stroke risk were estimated using Cox regression models. The analyses of smoking trajectory classes and of smoked pack-years and stroke risk were adjusted for sex, educational level, family history of strokes, leisure-time physical activity, BMI, alcohol consumption, and presence of hypertension, hypercholesterolemia, and diabetes. Starting age and ending age of smoking were studied only among ever-smokers (n = 8941) and the analyses were adjusted for smoked pack-years by age of 46 years to detect possible sensitive time period of smoking. The follow-up time in all Cox regression analyses started from birth. Subjects were censored at date of stroke, date of death, or end of study (31st of December 2015). IBM SPSS Statistics 24 were used for these statistical analyses.

## Results

The summed amount of years cohort members were followed up from their birth date until the end of follow-up period was in total 542,140 person-years. Model fit parameters of trajectory models with 1–7 classes are presented in Table 2. The seven-class model failed to converge. The six-class model provided the most appropriate interpretation of the data showing better fit than models with 1–5 classes and having sufficient number of participants in each class. In the six-class model, the first smoking trajectory class were 'non-smokers' (n = 3223, 26.9% of 11,999) and were used as the reference group in the analyses. Class 2 included 'quitters in their twenties' (mean smoking age 16–26 years, n = 2034, 17.0%), class 3 included 'quitters in their early thirties' (mean smoking age 16–31 years, n = 3356, 28.0%), class 4 included 'quitters in their middle thirties' (mean smoking age 17–34 years, n = 1128, 9.4%), class 5 included 'quitters in their forties' (mean smoking age 18–43 years, n = 961, 8.0%), and class 6 included 'continuing smokers' (n = 1297, 10.8%). The starting ages of smoking were similar in all trajectory classes, and the trajectory classification was based mainly on the ending ages of smoking (Fig 1).

In total 8941 (74.5%) cohort members had smoked at some point in their lives. Of them, 779 were considered as non-smokers in the smoking trajectory model, resulting in 8162 persons (68.0%) who had smoked regularly. Mean starting age of smoking was 16.5 years (standard error 0.21 years). The presence of stroke risk factors, means of smoked pack-years by age 31 and 46, means of smoked years, and means of starting and ending age of smoking in each trajectory class are shown in the Table 3. Logically, subjects with longer smoking history had more smoked pack-years than those with shorter history.

During the follow-up period, 352 (2.9%) persons had a stroke resulting in an incidence of 64.9/100,000 person-years (Table 4). The incidences were similar between women (65.4/

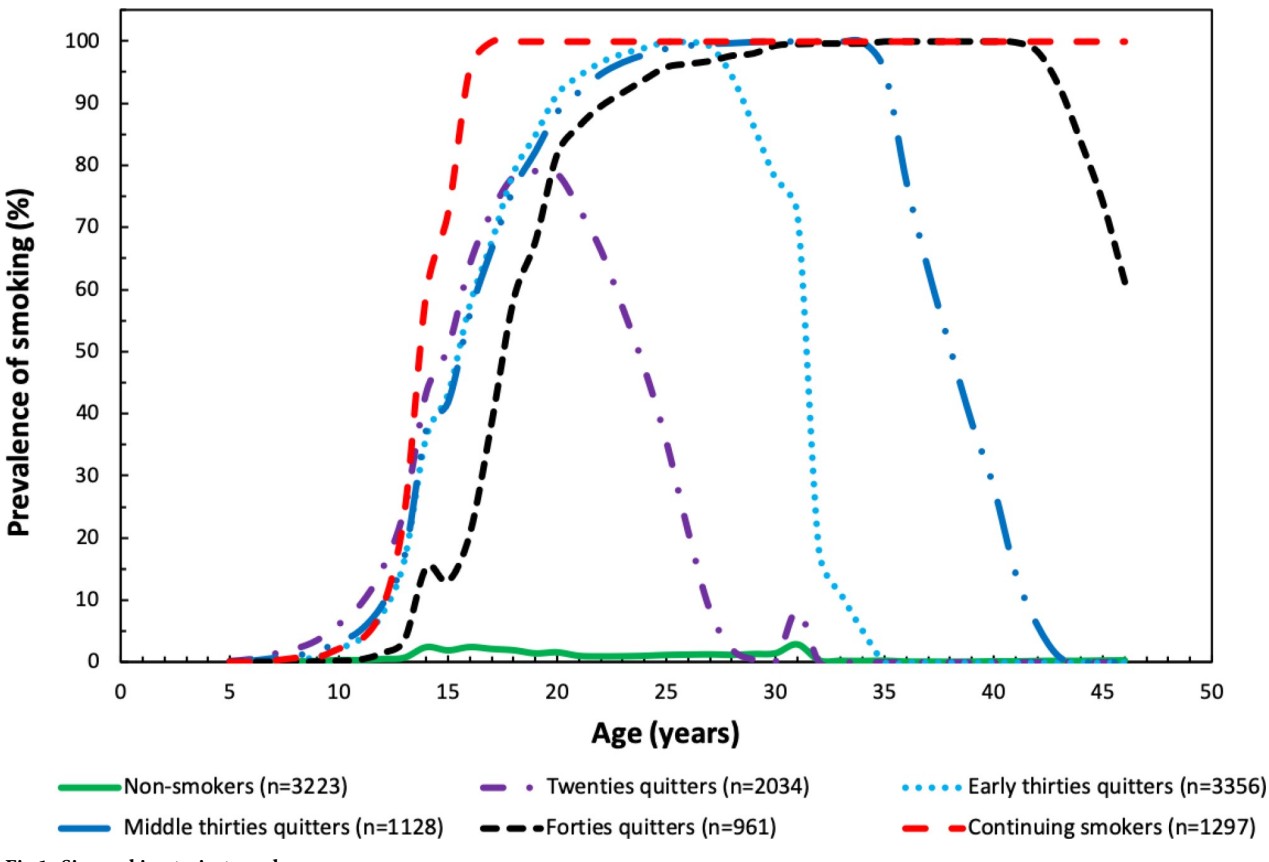

**Fig 1. Six smoking trajectory classes.**

100,000 person-years) and men (64.4/100,000 person-years). Of all strokes, 113 (32.1%) were permanent ischemic strokes, 108 (30.7%) were TIAs, 59 (16.8%) were SAHs, 26 (7.4%) were ICHs, and 46 (13.1%) were other cerebrovascular events. The mean age of stroke onset was different between ischemic stroke (42.5 years, SD 6.8) and haemorrhagic stroke (37.3 years, SD 11.4) with mean difference of 5.2 years (95% CI 2.64–7.86).

### Association between smoking habits and risk of stroke

The incidences and covariate-adjusted HR for any stroke, ischemic stroke, or haemorrhagic stroke according to smoking trajectory classes are shown in the Table 4. Continuing smokers had HR of 1.69 (95% CI 1.10–2.60) to have any kind of stroke when compared to non-smokers. The trajectory class 'quitters in their early thirties' had an increased risk of stroke (HR 1.44; 95% CI 0.94–2.20), in particular haemorrhagic stroke (HR 2.73; 95% CI 1.12–6.65) when compared to non-smokers.

Smoked pack-years were associated with risk of any stroke, ischemic stroke, and haemorrhagic stroke (Table 4). For every pack-year smoked before the age of 46, the risk of stroke was 1.04 times higher (95% CI 1.03–1.06). Starting or ending ages of smoking were not associated with stroke risk (HR 0.99; 95% CI 0.94–1.04 and HR 0.99; 95% CI 0.97–1.01, respectively).

### Discussion

In this study we found that a smoking trajectory of continuing smoking was associated with an increased risk of stroke and that a smoking trajectory of those who quit smoking in their early

**Table 3. Basic characteristics of six smoking trajectory classes according to 46-years follow-up.**

| | All (N = 11999) | Non-smokers (n = 3223) | Quitters in their twenties (n = 2034) | Quitters in their early thirties (n = 3356) | Quitters in their middle thirties (n = 1128) | Quitters in their forties (n = 961) | Continuing smokers (n = 1297) |
|---|---|---|---|---|---|---|---|
| | n (%) | n (%) | n (%) | n (%) | n (%) | n (%) | n (%) |
| Female sex | 5860 (48.8%) | 1870 (58.0%) | 1029 (50.6%) | 1447 (43.1%) | 464 (41.1%) | 430 (44.7%) | 622 (48.0%) |
| Basic education | 1094 (9.1%) | 182 (5.6%) | 199 (9.8%) | 461 (13.7%) | 98 (8.7%) | 49 (5.1%) | 104 (8.0%) |
| Secondary education | 8387 (69.0%) | 2040 (63.3%) | 1419 (69.8%) | 2365 (70.5%) | 817 (72.4%) | 727 (75.7%) | 1019 (78.6%) |
| Tertiary education | 2519 (21.0%) | 1001 (31.1%) | 417 (20.5%) | 530 (15.8%) | 213 (18.9%) | 185 (19.3%) | 174 (13.4%) |
| Family history of stroke | 1912 (15.9%) | 496 (15.4%) | 339 (16.7%) | 528 (15.7%) | 186 (16.5%) | 156 (16.2%) | 208 (16.0%) |
| Hypertension | 4769 (39.7%) | 1234 (38.3%) | 767 (37.7%) | 1355 (40.4%) | 476 (42.2%) | 396 (41.2%) | 542 (41.8%) |
| Hypercholesterolemia | 6926 (57.7%) | 1821 (56.5%) | 1169 (57.7%) | 2008 (59.8%) | 685 (60.7%) | 525 (54.6%) | 718 (55.4%) |
| Diabetes | 840 (7.0%) | 178 (5.5%) | 130 (6.4%) | 274 (8.2%) | 98 (8.7%) | 63 (6.6%) | 97 (7.5%) |
| | Mean (SE) | Mean (SE) | Mean (SE) | Mean (SE) | Mean (SE) | Mean (SE) | Mean (SE) |
| Physical activity (hours / month) | 33.7 (SE 0.88) | 36.2 (SE 0.78) | 35.0 (SE 1.31) | 33.1 (SE 1.71) | 33.4 (SE 1.52) | 31.9 (SE 1.06) | 28.7 (SE 0.86) |
| BMI (kg/m$^2$) | 27.1 (SE 0.06) | 26.6 (SE 0.10) | 27.0 (SE 0.14) | 27.3 (SE 0.11) | 27.7 (SE 0.18) | 27.2 (SE 0.16) | 27.1 (SE 0.14) |
| Alcohol consumption (g/ day) | 11.0 (SE 0.38) | 7.1 (SE 0.37) | 9.7 (SE 0.58) | 12.8 (SE 0.69) | 12.3 (SE 0.76) | 13.4 (SE 0.68) | 15.4 (SE 0.59) |
| Pack-years by age 31 | 4.3 (SE 0.09) | 0.5 (SE 0.09) | 3.0 (SE 0.19) | 5.9 (SE 0.19) | 5.7 (SE 0.35) | 5.9 (SE 0.28) | 9.0 (SE 0.22) |
| Pack-years by age 46 | 7.8 (SE 0.70) | 1.6 (SE 0.53) | 5.7 (SE 0.89) | 9.7 (SE 1.15) | 9.4 (SE 0.93) | 11.6 (SE 0.60) | 17.5 (SE 0.42) |
| Years smoked | 16.7 (SE 0.50) | 7.6 (SE 1.78) | 10.0 (SE 0.83) | 14.2 (SE 0.54) | 18.0 (SE 0.92) | 25.0 (SE 0.98) | 31.0 (SE 0.40) |
| Starting age | 16.5 (SE 0.21) | 18.9 (SE 0.50) | 16.1 (SE 0.26) | 16.4 (SE 0.19) | 16.5 (SE 0.30) | 18.4 (SE 0.19) | 14.6 (SE 0.07) |
| Ending age | 33.2 (SE 0.57) | 26.2 (SE 2.04) | 26.0 (SE 0.97) | 30.6 (SE 0.62) | 34.4 (SE 0.95) | 43.4 (SE 1.10) | 45.5 (SE 0.36) |

The characteristics are reported for imputed data with no missing values. SE = standard error, kg/m$^2$ = kilograms per square-meter, g/day = grams per day.

thirties was associated with increased risk for haemorrhagic stroke when compared to non-smokers. Further, the number of smoked pack-years was associated with risk of stroke. We did not find associations between smoking starting age or ending age and stroke risk.

The results of this study show that accumulation of smoked pack-years might be more crucial to stroke risk than starting or ending age of smoking. This suggests that the harmful effects of smoking depend on dose and duration of smoking and are irrespective of age when smoking occurred. In this present study, the intensity of smoking, e.g. daily cigarette consumption, was not studied separately from pack-years. A previous study examining young women found that not the duration of smoking but the dose of daily smoking and smoked pack-years were strongly associated with increased risk for ischemic stroke[16]. Another previous study also showed the association between smoking and risk of stroke to be dose-dependent[17]. The findings of this study and previous studies suggest that measuring pack-years may be optimal when assessing the smoking-related stroke risk. In a previous Finnish study comparing different ways to estimate longitudinal risk factors for cardiovascular disease mortality, a model representing lifetime accumulation of smoking had better predictive ability than a model using only the most recent measured information of smoking status[35].

In this study, smoking trajectory of continuing smoking was related to stroke risk, but also to higher amount of pack-years. High amount of pack-years increased the stroke risk, and this association might be irrespective of the smoking trajectory. We did not adjust analyses of smoking trajectories and stroke risk with pack-years to avoid multicollinearity. Other

**Table 4. Incidences and hazard ratios of stroke according to six smoking trajectory classes, pack-years, and smoking starting and ending age.**

| | Stroke | | | Ischemic stroke (including IS and TIA) | | | Haemorrhagic stroke (including SAH and ICH) | | |
|---|---|---|---|---|---|---|---|---|---|
| | No. of strokes | No. per 100,000 | HR (95% CI) | No. of strokes | No. per 100,000 | HR (95% CI) | No. of strokes | No. per 100,000 | HR (95% CI) |
| **All (N = 11999)** | | | | | | | | | |
| | 352 (2.9%) | 64.9 | n.a. | 221 (1.8%) | 40.8 | n.a. | 85 (0.7%) | 15.7 | n.a. |
| **Smoking exposure trajectory classes** | | | | | | | | | |
| Non-smokers (n = 3223) | 76 (2.4%) | 49.6 | ref. | 53 (1.6%) | 34.6 | ref. | 11 (0.3%) | 7.2 | ref. |
| Quitters in their twenties (n = 2034) | 52 (2.6%) | 58.8 | 1.14 (0.73–1.78) | 31 (1.5%) | 35.1 | 1.01 (0.57–1.80) | 13 (0.6%) | 14.7 | 1.90 (0.60–5.98) |
| Quitters in their early thirties (n = 3356) | 105 (3.1%) | 72.7 | 1.44 (0.94–2.20) | 64 (1.9%) | 44.3 | 1.30 (0.78–2.15) | 32 (1.0%) | 22.1 | 2.73 (1.12–6.65) |
| Quitters in their middle thirties (n = 1128) | 31 (2.7%) | 65.1 | 1.26 (0.65–2.44) | 18 (1.6%) | 37.8 | 1.05 (0.45–2.48) | 10 (0.9%) | 21.0 | 2.28 (0.66–7.89) |
| Quitters in their forties (n = 961) | 31 (3.2%) | 68.4 | 1.28 (0.73–2.24) | 18 (1.9%) | 39.7 | 1.10 (0.55–2.23) | 10 (1.0%) | 22.1 | 2.42 (0.83–7.02) |
| Continuing smokers (n = 1297) | 57 (4.4%) | 90.3 | 1.69 (1.10–2.60) | 38 (2.9%) | 60.2 | 1.73 (1.03–2.91) | 9 (0.7%) | 14.3 | 1.49 (0.50–4.45) |
| **Smoked pack-years** | | | | | | | | | |
| Per pack-year by 31 | n.a. | n.a. | 1.04 (1.02–1.06) | n.a. | n.a. | 1.03 (1.00–1.06) | n.a. | n.a. | 1.07 (1.04–1.11) |
| Per pack-year 31–46 | n.a. | n.a. | 1.07 (1.04–1.10) | n.a. | n.a. | 1.08 (1.04–1.12) | n.a. | n.a. | 1.06 (0.99–1.13) |
| Per pack-year by 46 | n.a. | n.a. | 1.04 (1.03–1.06) | n.a. | n.a. | 1.04 (1.03–1.06) | n.a. | n.a. | 1.04 (1.02–1.07) |
| **Starting and ending age of smoking** | | | | | | | | | |
| Starting age in years | n.a. | n.a. | 0.99 (0.94–1.04) | n.a. | n.a. | 1.01 (0.94–1.08) | n.a. | n.a. | 0.94 (0.85–1.05) |
| Ending age in years | n.a. | n.a. | 0.99 (0.97–1.01) | n.a. | n.a. | 0.99 (0.96–1.02) | n.a. | n.a. | 0.98 (0.94–1.02) |

Cox regression models of smoking exposure trajectory groups classes and of smoked pack-years were adjusted for sex, educational level, family history of strokes, physical activity, BMI, alcohol consumption, and presence of hypertension, hypercholesterolemia, and diabetes. Starting age and smoking ending age of smoking were studied only among ever-smokers and were adjusted for smoked pack-years by age of 46 years. Pack years, starting age, and ending of smoking were studied as continuous variables, and results are shown as HRs per pack year or increase of one year in starting or ending age. IS = ischemic stroke, TIA = transient ischemic attack, SAH = subarachnoid haemorrhage, ICH = intracerebral haemorrhage, No. = number, HR = hazard ratio, 95% CI = 95% confidence interval, n.a. = not applicable, ref = reference group.

trajectories than continuing smoking represented smoking histories where participants quit smoking during follow-up. It is known that stroke risk declines when an individual quits smoking[14,15]. Previous studies investigating the associations between smoking and preconditions for stroke risk have found that harmful arterial changes among those who quitted smoking reversed into similar levels to those who never smoked[36]. In the Bogalusa Heart Study, the duration of smoking years since childhood, but not smoking at age 8 to 17, was associated with changes in arterial thickness that can be considered as a preclinical marker to ischemic stroke risk[37]. Furthermore, a recent study showed that the effect of adolescence smoking on future stroke risk might be lower than expected from previous studies, due to previous studies' failure to follow the changes in smoking habits after baseline testing[38]. The current study was able to follow the changes in smoking habits from childhood to adulthood.

In the current study we also found that smoking trajectory of those who quitted smoking in their early thirties was associated with an increased risk for haemorrhagic stroke in particular.

This suggests that there might be sensitive period during adolescence and young adulthood when susceptibility for effects of smoking is greater than in later life and that the timing of smoking might play a role in development of haemorrhagic stroke risk. Nonetheless, it should be noted that mean age of onset of haemorrhagic strokes, in particular subarachnoid haemorrhages, is younger than that of ischemic strokes which might partly explain the finding[39–42]. Future studies are needed to investigate if early smoking associates to haemorrhagic strokes despite the smoked pack-years.

This current study had some limitations. The main limitation of this study was that the age intervals of follow-up questionnaires (14, 31, and 46 years) were rather long apart. Smoking status was asked only in these three time points and was estimated for the remaining age points that were used in smoking trajectory model (between 5 and 46 years) based on starting and ending age. To reduce recall bias due to gaps between follow-up questionnaires, the 31 years follow-up assessment was selected as the primary source for data on smoking starting age and the 46 years follow-up for smoking ending age.

Second, in this population most smokers started smoking around the same age. In the six-class smoking trajectory model the mean starting ages were similar in all classes. Therefore, the trajectories mainly represent the ending age and the duration of smoking, and no comparisons between classes with same duration but different starting and ending age of smoking could be made. Some trajectory classes, e.g. quitters in their early thirties and quitters in their middle thirties, were similar to each other with respect to starting and ending age of smoking and smoked pack years. This might challenge the clinical interpretation of the results.

As another limitation, the sample sizes of the stroke groups (e.g. the group of haemorraghic strokes) were rather small, which might have underpowered this study to detect differences. Stroke at young age is a rare phenomenon[42], and despite the relatively high incidence of in particularly subarachnoid haemorrhages in Finland[39,43], and the large number of person-years in this follow-up study, only 352 people had a stroke. Furthermore, due to the limited number of stroke cases, different subtypes of ischemic or haemorrhagic strokes were not studied separately. Previous studies have shown that smoking increases especially the risk of SAH and ischemic stroke but might not increase the risk of ICH[17,44–46]. Additionally, it should be noted that use of smokeless tobacco was not studied, though it might increase the stroke risk[47].

A further limitation of this study is the loss to follow-up in clinical and questionnaire surveys that assessed the smoking status and covariates. In addition to loss to follow-up of subjects, there were missing measurements of available subjects, and in total 27.5% of all original data were missing. The multiple imputation method was used in this study to complete the missing data of smoking statuses and covariates, as it reduces selection bias due to selective loss to follow-up, and also increases statistical power[48,49]. A previous study from the same birth cohort data has shown that high alcohol consumption, low educational level, unemployment, and being single at age 31 predicted lower participation at follow-up examination and questionnaires[50]. This might have affected the results if the multiple imputation was not used. However, it should be noted that the selection of variables in multiple imputation model might also affect the results[51].

The current study also had several strengths. First, it had three different approaches to study the association between smoking and stroke risk: 1) smoking trajectories, 2) smoked pack-years, and 3) starting and ending age of smoking. Second, a large and unselected population-based birth cohort was used in this study with over 500,000 person-years of follow-up. Results of this naturalistic real-world data set are highly generalizable to Finnish population. The data collection started from the second trimester of cohort members' antenatal period and follow-up lasted up to 50 years of age. The data collection was prospective reducing the

potential for information bias, and questionnaire and clinical examination data were combined with comprehensive nationwide registers. Information on stroke diagnoses were collected from nationwide registers that were complete and continuous without loss of follow-up. Third, the smoking status was measured at several age-points with several characteristics of smoking and the information on stroke diagnoses from nationwide registers was complete for the whole cohort. Finally, multiple imputation was conducted to reduce selection bias and loss of statistical power due to missing data.

## Conclusions

This study showed that accumulation of smoking history is associated with increased risk of stroke until age of 50 years. The increased stroke risk does not depend on the age at which smoking started. Given that the majority starts smoking at young age, primary prevention of strokes should focus on adolescent smoking.

## Supporting information

**S1 Table. Variables used in the multiple imputation procedure.**
(DOCX)

## Author Contributions

**Conceptualization:** Ina Rissanen, Markus Paananen, Jouko Miettunen, Mirjam I. Geerlings.

**Data curation:** Ina Rissanen, Jouko Miettunen.

**Formal analysis:** Ina Rissanen, Petteri Oura, Markus Paananen.

**Funding acquisition:** Jouko Miettunen, Mirjam I. Geerlings.

**Investigation:** Ina Rissanen, Mirjam I. Geerlings.

**Methodology:** Ina Rissanen, Petteri Oura, Markus Paananen, Mirjam I. Geerlings.

**Project administration:** Ina Rissanen, Mirjam I. Geerlings.

**Resources:** Ina Rissanen, Jouko Miettunen, Mirjam I. Geerlings.

**Software:** Ina Rissanen, Petteri Oura.

**Supervision:** Jouko Miettunen, Mirjam I. Geerlings.

**Validation:** Ina Rissanen, Petteri Oura.

**Visualization:** Ina Rissanen.

**Writing – original draft:** Ina Rissanen.

**Writing – review & editing:** Ina Rissanen, Petteri Oura, Markus Paananen, Jouko Miettunen, Mirjam I. Geerlings.

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
