## [Decision Letter · Decision Letter 0]

12 Sep 2019

PONE-D-19-20624

Smoking trajectories and risk of stroke until age of 50 years

– the Northern Finland Birth Cohort 1966

PLOS ONE

Dear Dr. Rissanen,

Thank you for submitting your manuscript to PLOS ONE. After careful consideration, we feel that it has merit but does not fully meet PLOS ONE’s publication criteria as it currently stands. Therefore, we invite you to submit a revised version of the manuscript that addresses the points raised during the review process.

We would appreciate receiving your revised manuscript by Oct 27 2019 11:59PM. To enhance the reproducibility of your results, we recommend that if applicable you deposit your laboratory protocols in protocols.io, where a protocol can be assigned its own identifier (DOI) such that it can be cited independently in the future. For instructions see: http://journals.plos.org/plosone/s/submission-guidelines#loc-laboratory-protocols

We look forward to receiving your revised manuscript.

Kind regards,

Thomas Behrens

Academic Editor

PLOS ONE

Journal Requirements:

Additional Editor Comments (if provided):

In addition to our reviewer’s comments, there are several issues that need to be discussed and elaborated in more detail before publication can be granted.

Major:

1. As also noted by our reviewer, the biggest limitation of your work lies in the long gaps of 16 years between the different surveys time points. This needs to be critically discussed in more detail.

2. Further limitations to be discussed include the limited response at the last survey (56% of the original cohort), which may also create selection effects. Are the cohort members representative for the population, i.e. is the high smoking prevalence also seen in the general population? The authors should consider to introduce population weights to approach representativeness.

3. Item response is also a severe limitation, as the authors note (27.5%). Multiple imputation has been used, but the authors do not report results without application of imputation. Please note that the abbreviation MI for multiple imputation may be misleading (myocardial infarction) and should therefore be avoided.

4. The number of pack-years by age 46 appears to be rather low (15.8 pack-years on average), given that subjects started early (at 16 years of age). Is this number plausible/representative for the Finnish population?

5. I do not agree with the notion that a non-significant result indicates “no association”. Most trajectories show positive associations, which fail to demonstrate statistical significance though due to the limited power (as the authors acknowledge). Interpretation of “significance” as an implied “truth” when associations of 1.4 or more are considered lacking and HRs of 1.08 for pack-years are seen as important, is a severe distortion of the scientific process. Similarly, late quitters apparently do not indicate an association (HR 1.08, or <1), whereas risk per pack-years by age 46 years show 1.04, but with statistical significance?

6. Some analyses in Tab. 3 need to be clarified: Cumulative exposure:

- Risk by pack-year before age 31 years? What was analysed?

- Starting-Ending age? What was modelled?

Minor:

Abstract: Please state confounders explicitly.

Methods: Please state how educational level was classified in more detail (years of education, which degrees?)

How was PA measured?

Results: Model fit parameters should be reported.

Fig. Axes require description.

Tables:

- Tab. 2: Indicate all units in brackets. What is reported: means, medians?

- What is the criterion to be coded as non-smoker? NS reported on average 1.5 pack-years by age 46 years. Please add definition in Methods.

- Please add footnotes for all abbreviations used.

References: Some references are not according to PlosOne style (e.g., journal titles of Ref. 21, 22, 29, and 33 need to be abbreviated. Please add dois for all references.

Reviewers' comments:

Reviewer's Responses to Questions

**Comments to the Author**

1. Is the manuscript technically sound, and do the data support the conclusions?

Reviewer #1: Partly

2. Has the statistical analysis been performed appropriately and rigorously? 

Reviewer #1: No

3. Have the authors made all data underlying the findings in their manuscript fully available?

Reviewer #1: No

4. Is the manuscript presented in an intelligible fashion and written in standard English?

Reviewer #1: Yes

5. Review Comments to the Author

Reviewer #1: Here you can find the overall impression I got from the manuscript (MS). The line by line comments you can find as an attachment.

Decision: Few major revisions & several minor revisions.

The strength of the study is that it has three different approaches to study the association between smoking and stroke risk. It studies the association between smoking trajectories, cumulative smoking exposure and starting and ending age of smoking with stroke risk. As the scientific studies that examine heterogeneous trajectory patterns of smoking using finite mixture models has only started to accumulate, this paper is timely.

However, conducting the trajectory modelling and especially reporting about the statistical analyses in the Results are not described in sufficient detail. The reader does not get all the needed information for evaluating the identification of trajectories and interpreting the sensitivity analyses.

The discussion could be more profound. For example, smoking has been studied in three time points (which is the minimum demand for identifying trajectories and therefore acceptable number), but the age points are rather long apart. Because of this, some smoking patterns may not be observed. Naturally, this matter cannot be fixed afterwards, but it could be discussed. Also, the paragraphs in the Discussion are partly disconnected.

Many statements and terms are unclear and terms need to be defined better / earlier in the MS. Additionally, the use of references is not precise in many parts, and references are missing. I suggest to rethink which references to put in the Introduction and which in the Discussion in order to have more fluent story and to avoid repetition.

The amount of my suggestions may give the expression that there are lot of revisions to do. Please, do not worry: most of my comments are minor ones. Even if I happened to notice misspelling, I corrected that too - just to help to improve your MS. Nonetheless, this MS could still be revised in language.

6. PLOS authors have the option to publish the peer review history of their article (what does this mean?). If published, this will include your full peer review and any attached files.

Reviewer #1: No

---

## [Author Response · Author response to Decision Letter 0]

17 Oct 2019

October 5th, 2019 

Dear Academic Editor Thomas Behrens, 

Please find enclosed our revised manuscript entitled “Smoking exposure trajectories and risk of stroke until age of 50 years – the Northern Finland Birth Cohort 1966” (PONE-D-19-20624).

We thank you and the reviewer for the comments and suggestions to improve our manuscript. Below, we present our detailed response to each comment. The original question/request is marked with plain font followed by our response with italic font. We have revised the manuscript according to both your and reviewer’s comments (changes in the manuscript are marked with the ‘track changes’ tool).

We hope that we have satisfactorily responded to the constructive critique and believe those changes have markedly improved our manuscript. We hope that our manuscript could be now accepted for publication.

Yours sincerely,

Ina Rissanen, MD, Ph.D.

Corresponding author

Departments of Neurosurgery and Neurology, Oulu University Hospital,

PL 10, 90029 OYS, Oulu, Finland

and

Julius Center for Health Sciences and Primary Care, University Medical Center Utrecht and Utrecht University,

Huispost nr. STR 6.131, P.O. Box 85500, 3508 GA Utrecht, The Netherlands

Telephone +31618505066

Email ina.rissanen@gmail.com

ORCID 0000-0002-6869-0437 

Smoking exposure trajectories and risk of stroke until age of 50 years – the Northern Finland Birth Cohort 1966 (PONE-D-19-20624)

Additional Editor Comments:

Major:

1. As also noted by our reviewer, the biggest limitation of your work lies in the long gaps of 16 years between the different surveys time points. This needs to be critically discussed in more detail.

Thank you for pointing out this critical issue. We agree that this is a limitation of this study. We have now discussed this in more detail in limitation section of the study. We acknowledge that it is a major limitation of this study that smoking status was assessed only in three time points with questionnaires and these time points were long apart. However, information on starting and ending age of smoking was used to estimate the smoking status for each year in the gap periods. In addition, stroke outcomes were available for each day of cohort members’ lives from national registers that consist of longitudinal data of each individual.

2. Further limitations to be discussed include the limited response at the last survey (56% of the original cohort), which may also create selection effects. Are the cohort members representative for the population, i.e. is the high smoking prevalence also seen in the general population? The authors should consider to introduce population weights to approach representativeness.

Thank you for pointing out this issue. We have now added a new paragraph to the discussion section to address this limitation. The loss of follow-up is a problem in this study, as is the case for all prospective cohort studies. However, to reduce the selection effects, we have conducted the multiple imputation procedure. In multiple imputation, information on previous surveys was used to impute the variables of last survey.

Northern Finland Birth Cohort 1966 is an unselected and naturalistic cohort that includes 96.3 per cent of all births in the two Northernmost provinces in Finland (Oulu and Lapland) during 1966. Therefore, it is representative of the general population of Finland with the same age. Smoking prevalences of this birth cohort are similar to other population of Finland (https://thl.fi/en/web/alcohol-tobacco-and-addictions/tobacco/smoking-in-finland). Northern Finland Birth Cohort 1966 (https://www.oulu.fi/nfbc/node/44315) has similar characteristics as the general Finnish population (ethnicity, education, sex, rural/urban place of residence etc.) and loss of follow-up was taken into account with multiple imputation. Because of this, we do not think it is necessary to introduce population weights.

3. Item response is also a severe limitation, as the authors note (27.5%). Multiple imputation has been used, but the authors do not report results without application of imputation. Please note that the abbreviation MI for multiple imputation may be misleading (myocardial infarction) and should therefore be avoided.

We agree that a limitation of this study is the presence of missing data both due to missing items and loss of follow-up (in total 27.5% of data missing). However, we reported the results of the complete case analyses (results without application of multiple imputation) as sensitivity analyses in the results section. We have now added more discussion about the differences between results of imputed data and complete case analyses. We have removed the abbreviation MI from the manuscript as it was misleading.

4. The number of pack-years by age 46 appears to be rather low (15.8 pack-years on average), given that subjects started early (at 16 years of age). Is this number plausible/representative for the Finnish population?

Thank you for this notion. We added a new line to Table 3 to show also the average length of smoking exposure in years in each trajectory class. The mean number of pack-years seems to be approximately half of the mean duration of smoking years which suggests that the average daily smoking dose is around 10 cigarettes in this cohort. There are no previous studies reporting the average amount of smoked pack-years among Finnish individuals aged 50 years. However, the Eurostat statistics of European Union (https://ec.europa.eu/eurostat/statistics-explained/index.php/Tobacco_consumption_statistics) report that consuming more than 20 cigarettes per day is very rare in Finland. These statistics indicate that nearly all smokers in Finland consume less than 20 cigarettes daily and suggest that the mean daily consumption in this study is representative of Finnish population.

5. I do not agree with the notion that a non-significant result indicates “no association”. Most trajectories show positive associations, which fail to demonstrate statistical significance though due to the limited power (as the authors acknowledge). Interpretation of “significance” as an implied “truth” when associations of 1.4 or more are considered lacking and HRs of 1.08 for pack-years are seen as important, is a severe distortion of the scientific process. Similarly, late quitters apparently do not indicate an association (HR 1.08, or <1), whereas risk per pack-years by age 46 years show 1.04, but with statistical significance?

We agree with the editor concerning the problems with interpretation of “significance” in our manuscript. We have tried to clarify the interpretation of the results in Results and Discussion sections. Because the pack-years are measured as continuous variable and smoking exposure trajectories are nominal variables, the interpretation of these results are not comparable. Hazard ratio for stroke per pack-year by age 46 is 1.04 which might not seem as clinically relevant as higher HRs of smoking trajectory classes. Nonetheless, if an individual had for example 16 pack-years, the risk for stroke would be 1.04 to the power of 16, i.e., 1.87. This is similar to HR for stroke among the smoking exposure trajectory class ‘continuous smokers’, and much higher than the HR of 1.08 of class ‘late starters’.

It is true that non-significant results do not mean that there would not be any “true” association. Therefore, we have removed such claims from the manuscript. 

6. Some analyses in Tab. 3 need to be clarified: Cumulative exposure:

- Risk by pack-year before age 31 years? What was analysed?

- Starting-Ending age? What was modelled?

We apologize for the unclear presentation of the results. We have now clarified these in the manuscript. Pack years, starting age, and ending of smoking were studied as continuous variables in Cox regression model of any stroke, ischemic stroke, or haemorrhagic stroke. The results are shown as hazard ratios per pack year increase or increase of one year in starting or ending age. Starting age and smoking ending age of smoking were studied only among ever-smokers and were adjusted for smoked pack-years by age 46 years.

Minor:

7. Abstract: Please state confounders explicitly.

We have now stated the confounders explicitly in the abstract.

8. Methods: Please state how educational level was classified in more detail (years of education, which degrees?)

We have clarified the classification of educational level in the Methods section.

9. How was PA measured?

We have added information on how physical activity was measured.

10. Results: Model fit parameters should be reported.

We have added Table 2 to Results section to report the model fit parameters. 

11. Fig. Axes require description.

We have edited the Figure 1. We added description for axes.

12. Tab. 2: Indicate all units in brackets. What is reported: means, medians? Please add footnotes for all abbreviations used.

We added footnotes for all abbreviations used and indicated units in brackets.

13. Tab. 2: What is the criterion to be coded as non-smoker? NS reported on average 1.5 pack-years by age 46 years. Please add definition in Methods.

The smoking exposure trajectory class ‘non-smokers’ was formed in trajectory analysis, and therefore, some may have minor smoking history during their lifetime. We acknowledge that this is confusing, and we have tried to clarify this in the manuscript. We added a definition in Methods section.

14. References: Some references are not according to PlosOne style (e.g., journal titles of Ref. 21, 22, 29, and 33 need to be abbreviated. Please add dois for all references.

We have changed the references to be in PlosOne style with DOIs.

 

Reviewers' comments:

The strength of the study is that it has three different approaches to study the association between smoking and stroke risk. It studies the association between smoking trajectories, cumulative smoking exposure and starting and ending age of smoking with stroke risk. As the scientific studies that examine heterogeneous trajectory patterns of smoking using finite mixture models has only started to accumulate, this paper is timely.

However, conducting the trajectory modelling and especially reporting about the statistical analyses in the Results are not described in sufficient detail. The reader does not get all the needed information for evaluating the identification of trajectories and interpreting the sensitivity analyses.

The discussion could be more profound. For example, smoking has been studied in three time points (which is the minimum demand for identifying trajectories and therefore acceptable number), but the age points are rather long apart. Because of this, some smoking patterns may not be observed. Naturally, this matter cannot be fixed afterwards, but it could be discussed. Also, the paragraphs in the Discussion are partly disconnected.

Many statements and terms are unclear and terms need to be defined better / earlier in the MS. Additionally, the use of references is not precise in many parts, and references are missing. I suggest to rethink which references to put in the Introduction and which in the Discussion in order to have more fluent story and to avoid repetition.

The amount of my suggestions may give the expression that there are lot of revisions to do. Please, do not worry: most of my comments are minor ones. Even if I happened to notice misspelling, I corrected that too - just to help to improve your MS. Nonetheless, this MS could still be revised in language.

First, we would like to thank the reviewer of the evident investment of time and effort in reviewing our manuscript. All the authors of this manuscript acknowledge that we received most valuable comments not only for this manuscript, but also concerning the future research. We believe that these excellent comments from reviewer improved the manuscript greatly.

We agree that the reporting of trajectory analyses was insufficient, and we have now improved the reporting. We have added a new Table 2 to show the model fit parameters of trajectory analyses with 1-6 classes. We have also improved the main text of the manuscript in methods and results sections to provide more information for evaluating the identification of trajectories. We have also added more discussion about interpretation of the sensitivity analyses.

As the reviewer pointed out, one of the main limitations of this study is that smoking has been studied in three time point and they are rather long apart. This matter has now been discussed in more detail. However, it should be noted that in the trajectory analyses we used information on smoking status of each year of the follow-up. That is, we calculated smoking status for each age between ages 5 and 50 based on reported starting and ending age of smoking. Therefore, in trajectory analyses there were 45 time points based on which the trajectory classes were identified.

We have re-organized the discussion section. We believe that the paragraphs in the discussion are now better connected and the story is more fluent. We have made changes also to introduction section to avoid repetition. We agree that many statements and terms were unclear in the manuscript. We have now defined them better. We have added missing references.

We apologize the language errors of the manuscript as none of the authors is a native English speaker. We have revised the manuscript in language together with other revisions suggested by the reviewer.

Answers to the line-by-line comments are below.

 

Title page:

Line 4: It says that the author Ina Rissanen is part of the Department of Neurology and Neurosurgery and Medical Research Center Oulu (lines 7-9). However, in the submission data it is stated that Ina Rissanen is from the Universitair Medisch Centrum Utrecht.

Thank you for pointing out this error in the affiliations. We have now corrected the error.

Abstract:

Line 24: I would suggest to use a plural with “trajectory of smoking exposure” (so trajectories of…). After all, your aim is to study how the distinct smoking trajectories are related to stroke risk – not to identify a single growth trajectory that assumes the individuals to come from a single population and can be described and approximated with one trajectory.

We thank the reviewer for this comment. We have corrected the terminology in the manuscript.

Line 24: “Life course” is spelled without the hyphen. Correct throughout the MS.

Thank you for pointing out this typing error.

Line 25: “Aimed” should be written in small letters.

Thank you for pointing out this typing error.

Line 25: The sentence would be clearer if you would use the term relationship / association here as well as you did in the background description.

We have added the word “association” to this sentence.

Lines 24–25: I would like to see you use the same term of the trajectories throughout the MS (either smoking exposure trajectories / trajectories of smoking exposure).

We thank the reviewer for this comment. We decided to use the term “smoking exposure trajectories” throughout the manuscript.

Line 28: Comma after the 1966 or rephrasing the sentence. It is difficult to read when two numbers are following each other.

We have added comma after “1966”.

Lines 29–30: You also gathered data on smoking status at three age points. Mention that as well (or simply combine the smoking variables by stating that smoking behavior was assessed at ages 14, 31 and 46 years).

We combined the smoking variables in this sentence as suggested.

Lines 28–31: There is nothing about the statistical methods in the abstract. If the word count do not limit you too much, you might add a phrase on that. 

Thank you for this notion. We have now added a sentence about the statistical methods to the abstract.

Line 30: Do not use abbreviations (ICD) in abstract.

We have now removed this abbreviation.

Line 33: Comma before the 352 or rephrasing the sentence.

We have added comma after “352”.

Line 34: Put the statistical analyses in Methods, not in Results.

We thank the reviewer for pointing this out. We moved the part considering statistical analyses to the Methods section.

Lines 29–30, 39: It is not clear in the abstract nor yet in the methods of the MS what you mean by cumulative smoking and what are smoked pack-years. Please clarify the terms you use.

Thank you for this comment. After considering the terminology, we decided to delete the term “cumulative smoking exposure” entirely because it might be misleading. Therefore, we use only the term “pack-years” in the manuscript.

Introduction:

Line 43: According to the reference number 1, I believe that stroke is the second largest cause of death, but the third for disability. Please check the reference again and be precise.

We thank the reviewer for noticing this. Indeed, the reference number 1 stated that stroke is third largest cause of disability. We checked the reference and corrected this into the manuscript.

Line 45: You use the terms “young people”, “younger age” and “young age” in your MS. Do you always refer to people under 50 years (like in line 45) with these terms? Generally, it feels strange to use the term “young” when referring also to middle-aged people – even though I understand that the term “young” is used here since this is a young age to have a stroke. Nonetheless, please define somewhere (in the introduction or in the methods) what exact age you mean by “young” in the whole MS, use the same term throughout the MS, or consider using some other term (e.g., people under 50 years of age / young adults) to avoid confusion.

We thank the reviewer for pointing this out. We are now using the actual age ranges throughout the manuscript to avoid confusion and vague terms such as “young people”.

Lines 45–48: Divide to two sentences from “however” onward. I suggest to do the same with many other long sentences (e.g., in lines 49–51, 55–58, 58–60). Check this throughout the text. Just to make the text more fluent, use also some other term than “however” to avoid repetition (e.g., nonetheless).

We have divided these long sentences into two and tried to avoid repetition of word “however” in the text.

Line 53: Again, I would wish precision with terms. What does “later in life” and “younger age” mean here? Could you give specific age?

We changed these vague terms into specific ages.

Lines 58–59: The references to these “Most previous studies” are missing. Additionally, there are previous studies that have already identified diverse smoking trajectories, e.g.:

- Artaud et al. Trajectories of unhealthy behaviors in midlife and risk of disability at older ages in the Whitehall II Cohort Study. The Journals of Gerontology 2016, 71 (11), 1500-1506.

- Brook et al. Developmental trajectories of cigarette smoking from adolescence to the early thirties: personality and behavioral risk factors. Nicotine & Tobacco Research 2008, 10(8), 1283–1291.

- Chassin et al. Multiple trajectories of cigarette smoking and the intergenerational transmission of smoking: a multigenerational, longitudinal study of a Midwestern community sample. Health Psychology 2008, 27(6), 819-828.

- Salin et al. Smoking and physical activity trajectories from childhood to midlife. Int. J. Environ. Res. Public Health 2019, 16(6), 974.

Please acknowledge the use of trajectory modelling in smoking studies. I suggest for you to use these findings either in the introduction or discuss the similarities and differences of the findings of other trajectory studies and compare them to your own findings in the Discussion.

We thank the reviewer for offering us these references. We have read these papers and added comparison of their findings in the manuscript. However, we do not feel that discussing in more detail about similarities and differences of these findings would improve the manuscript, because the main focus of this study is the stroke risk. We are preparing another manuscript considering the characteristics of different smoking exposure trajectories, and we will discuss these aspects in that manuscript.

Line 60: This is a bit difficult phrase to understand: “…they have different patterns of future smoking habits”. Could it just be “…they have diverse / distinct /differing smoking patterns”?

We changed this to be “they have diverse smoking patterns” as suggested by the reviewer.

Lines 62–64: The reasoning for this study in lines 63-64 is good. However, references are missing again. What are the “most previous studies”? And how short is the “short follow-up”? Be more specific, please. 

We added references and defined the length of short follow-up to be less than 50 years.

Line 66–67: Why not simply “…the effect of smoking on stroke risk depended on…”?

We corrected this sentence as suggested.

Line 69: Are the words “…effect of smoking to stroke risk…” missing here? Additionally, do you mean “during adolescence” or would “starting from adolescence” be more accurate in this context?

We did the corrections that the reviewer suggested.

Line 74: Define “cumulative smoking” here or in the Methods (preferably the first time you mention it).

As mentioned earlier, we have changed the term “cumulative smoking” to be “pack-years” for clarification.

Line 77: Would it be possible to use the exact age here (not the term “childhood”)?

We replaced the term “childhood” with the exact age.

Methods:

Lines 82–105: I suggest you to give a subheading (3.1) for these paragraphs as well (e.g., Design and population of the study). 

We added a subheading as advised.

Line 88: If accessible, give information of reasons why participants declined use of their data, and what were the reasons for non-participation later in the study (loss to follow-up).

Unfortunately, no information on reasons why participants denied use of their data were available. We have stated this in the manuscript as well.

Line 92: Please state the average follow-up time per participant in the current study.

We added the mean follow-up time to the manuscript.

Line 92: Add L to “health”.

Thank you for pointing out this typing error.

Lines 98–102: If possible, divide to two sentences (long and difficult to understand).

We divided this sentence in two.

Line 102: Would it be possible to mention the registers’ names here since you talk about them here for the first time?

We added the names of the registers.

Lines 107–114: It is not stated clearly whether the smoking variable is binary at all three age points (14, 31, and 46 years). It seems it’s binary at the age of 14 and has three categories at the age of 31 and 46. However, on the line 174, you state that the smoking variables are binary.

We clarified this in the text. The smoking status was binary in all time points between 5 and 50 years.

Line 119: I suggest you to define in here the terms smoked pack years and cumulative smoking.

As mentioned earlier, we have changed the term “cumulative smoking” to be “pack-years” for clarification.

Line 123: This is the first time you use TIA. Please write the whole term open before using abbreviations.

We did the suggested correction.

Lines 127–128: The reference is missing.

We added references for this sentence.

Line 138: State in the 3.3 section the age points of the covariates.

We added information on the age points.

Lines 139–140: In the text you use terms low, moderate and high education and in the table 2 terms basic, secondary and tertiary. Be consistent throughout the text. Additionally, define what these education levels mean.

We corrected the terms to be consistent in the manuscript. Also, we added definitions to these in the method section.

Lines 140–141: The amount of physical activity seems rather high in table 2. Please define here more accurately how physical activity is assessed (e.g., does it include intensity, frequency, organized and unorganized activity, commuting, household work, leisure-time physical activity…).

The assessment of physical activity has been defined more accurately in the methods section.

Line 160: You use the term outcome on this line but it has not been defined in the text yet. It would be good to actually state somewhere that smoking is the exposure and stroke is the outcome.

We added a statement that smoking is the exposure and stroke is the outcome to the text.

Line 164: It is not necessary to write an abbreviation (here MAR) if you will not use it again in the text. Check this with all the abbreviations used.

The abbreviation was deleted.

Line 169: It is good that you made sensitivity analysis. However, the supplement 2 is not attached to the submission. Please add it.

We apologize this error in text. After revising our manuscript, we decided to include the information of supplement 2 in the main text of the manuscript, and therefore, it is not included in the supplement.

Line 173: Just to avoid confusion, I suggest not to use the term “cluster” with trajectory modelling since cluster analysis is a different type of statistical method. I suggest the use of trajectory, group, subgroup or class.

We changed the term “cluster” to be “class”.

Line 175: (This is a side note for possible future trajectory modelling.) Usually binary variables are not used in trajectory modelling in order not to lose valuable information. Thus, if you would have a variable of smoking with wider scale (e.g., how many cigarettes does the participant smoke in a day), it might be more suitable for trajectory modelling. Nonetheless, you have solved the problem by using logit-based model which is acceptable.

Thank you for this advice. In future, we will try to identify trajectories also with wider scale variables.

Line 175: It is recommended that also the model with one trajectory class is tested and reported (see Schoot et al. The GRoLTS Checklist: Guidelines for Reporting on Latent Trajectory Studies. Structural Equation Modeling: A Multidisciplinary Journal 2017, 24: 451–467.). Thus, read carefully this paper and then improve your reporting of the statistics and results concerning smoking trajectories. Try to report all the points mentioned in the Table 1 (Final List of Items of the Guidelines for Reporting on Latent Trajectory Studies) in the Schoot’s et al. article. Additionally, a table with the information from all the tested trajectory models (models 1–6) is needed in the Results. An example of this table (Table 2: Example Table With Hypothetical Results) is found in the above-mentioned article. It is important that the reader has the possibility to see the information criterion values, entropy values, average posterior probabilities and sample sizes per class of the different models in order to evaluate what is the importance given to the clinical significance of the modelling and what for the fit indices when choosing the final model.

We thank the reviewer for this essential criticism regarding the selection and reporting of trajectory models. We agree that also the one-class model should be tested, and the manuscript has now been supplemented accordingly. We have carefully read the guideline article referenced by the reviewer. Please find below the completed GRoLTS Checklist. We have also added a table summarizing all the tested trajectory models. However, as we used the PROC TRAJ (and not e.g. Mplus) to perform the trajectory analysis, some parameters were not available to us (BLRT, VLMR, entropy); instead, we used alternative parameters suggested by the developers of PROC TRAJ (in Jones et al., 2001) for model selection. We have therefore edited the GRoLTS table layout accordingly. To choose among the estimated trajectory models, we compared the BIC and AIC values, the Bayes factor, the estimated probabilities of group membership, class sizes, and clinical relevance of the classes relative to the aim of the study.

The GRoLTS Checklist:

1. Is the metric of time used in the statistical model reported? YES

2. Is information presented about the mean and variance of time within a wave? N/A (coeval cohort population with questionnaires at the ages of 14, 31 and 46)

3a. Is the missing data mechanism reported? N/A (no missing data in trajectory analysis)

3b. Is a description provided of what variables are related to attrition/missing data? N/A (no missing data in trajectory analysis)

3c. Is a description provided of how missing data in the analyses were dealt with? N/A (no missing data in trajectory analysis)

4. Is information about the distribution of the observed variables included? YES

5. Is the software mentioned? YES

6a. Are alternative specifications of within-class heterogeneity considered (e.g., LGCA vs. LGMM) and clearly documented? If not, was sufficient justification provided as to eliminate certain specifications from consideration? N/A (please see Jones et al, (2001))

6b. Are alternative specifications of the between-class differences in variance–covariance matrix structure considered and clearly documented? If not, was sufficient justification provided as to eliminate certain specifications from consideration? N/A (please see Jones et al, (2001)) 

7. Are alternative shape/functional forms of the trajectories described? YES

8. If covariates have been used, can analyses still be replicated? N/A (no covariates in trajectory analysis)

9. Is information reported about the number of random start values and final iterations included? NO (available from the authors if needed) 

10. Are the model comparison (and selection) tools described from a statistical perspective? YES 

11. Are the total number of fitted models reported, including a one-class solution? YES

12. Are the number of cases per class reported for each model (absolute sample size, or proportion)? YES

13. If classification of cases in a trajectory is the goal, is entropy reported? NO (not available in PROC TRAJ)

14a. Is a plot included with the estimated mean trajectories of the final solution? YES 

14b. Are plots included with the estimated mean trajectories for each model? NO (available from the authors if needed)

14c. Is a plot included of the combination of estimated means of the final model and the observed individual trajectories split out for each latent class? NO (available from the authors if needed)

15. Are characteristics of the final class solution numerically described (i.e., means, SD/SE, n, CI, etc.)? YES 

16. Are the syntax files available (either in the appendix, supplementary materials, or from the authors)? YES (available from the authors if needed)

Line 186: Since the class membership might change slightly when adding outcomes, predictors, covariates, etc. to the model, it would be more advisable to do all the analyses within the same modelling meaning the identification of trajectories and, in your case, the outcome (stroke). This would be possible by using Mplus. Unfortunately, I am not that familiar with SAS so I cannot say if you could do all these analyses by using SAS (and not SPSS). However, I think your solution is acceptable – especially if the posterior probabilities of the classes are high enough in all of the five classes and you will report them in the Results.

We thank the reviewer of this comment. We considered this when we planned this study. Unfortunately, none of us was familiar with the Mplus procedure and we decided to conduct the analyses with SAS and SPSS. In SAS PROC TRAJ application one cannot add the outcome in the model. We hope that in future we would gain more knowledge about Mplus trajectory modelling so we could truly compare these methods.

Line 189: Add Supplement 1?

We added the number 1.

Line 189: Change to “starting and ending age”?

We changed the sentence as suggested by the reviewer.

Line 190-191: Why adjusted only for smoked pack-years?

Can we find the response to this comment somewhere? 

Results:

Line 196: Define person years.

We added definition to person years.

Line 197: Add “smoking” and shorten the end of the sentence to “…five different smoking patterns were detected”.

We changed the sentence as suggested by the reviewer.

Line 202: Could you consider deleting the part “by themselves at least”? Gives an impression that these persons have only smoked alone.

We changed the sentence as suggested by the reviewer.

Line 203: I would change “tried smoking” to “smoked”. This gives an impression that one should really try to be a smoker.

We changed the sentence as suggested by the reviewer.

Line 203: How much is “very short”? Less than a month? Less than a year? Be more specific.

We changed this sentence to be more specific.

Line 204: “Ever smoker” sounds odd. Could it be “…who had smoked regularly”? Or what you are looking for here?

We changed the sentence as suggested by the reviewer.

Line 214: The abbreviation TIA has been used previously in line 123. There is no need to write it open here (but you need to write it in line 123).

This correction was made.

Lines 215–217: Would it be possible to show whether the ages (42.5 and 37.3) differ significantly from one another? You discuss a bit of the matter in the Discussion which is informative. However, could you be more profound in Discussion (e.g., what does previous literature say about why the onset of ischemic stroke comes later than onset of haemorrhagic stroke? This matter has apparently been studied in reference number 5. Please discuss whether / how your findings support these previous findings.)

We did independent samples t-test to show that these ages differ significantly from one another. However, as the focus of this study is in smoking exposure trajectories and stroke risk, we did not discuss this basic characteristic of stroke population very deeply in this manuscript.

Line 221: This heading could be misunderstood. It could mean that you talk about smoking risk and stroke risk. I believe you want to point out how smoking is a risk factor for stroke. Please rephrase.

We rephrased the heading as suggested by the reviewer.

Line 224: When compared to who?

“When compared to non-smokers” was added to this sentence.

Line 226: “…an increased risk…”

Thank you for pointing out this typing error.

Line 231: Define cumulative smoking earlier in the MS.

As mentioned earlier, we have changed the term “cumulative smoking” to be “pack-years” for clarification.

Line 241: The HR of haemorrhagic stroke are rather different than in the original analyses. Additionally, the CI is rather high - probably due to low number of strokes. It’s good that you talk about this limitation in the Discussion. However, please also discuss about the differences found between results of sensitivity analyses and original results in the Discussion.

We have added discussion about this topic. The main reason for this difference between imputed analyses and sensitivity analyses seems to be the low number of strokes in complete case analyses.

Line 245: Mark reference to the supplement material 2 (I assume you will add it to the MS)?

We added the information that was intended to be in Supplement 2 in the main text of the manuscript. Therefore, we did not add Supplement 2.

Discussion:

Lines 249–322: The first paragraph concerning the main findings is concise. However, usually the strengths and the limitations of the study are discussed in the end of Discussion. Thus, the second and third (and maybe even the fourth) paragraphs are in a bit unconventional place. Also, the Discussion between these results and previous findings could be more profound. Would it be possible, for example, to discuss more about the meaning of this study when compared to previous findings in this area (e.g., smoking trajectories from childhood to adulthood, associations of smoking and stroke in other high-income countries, etc.).

We would like to thank the reviewer of this critique. We have made some major changes to the structure and contents of the discussion section. We moved the limitations and strengths to the end of this section. We have added new discussion about the meaning of this study. We believe that the discussion is now more comprehensive.

Line 249: “…an increased…”

Thank you for pointing out this typing error.

Line 250: “…early quitters of smoking…”

We changed the sentence as suggested by the reviewer.

Line 251: “… stroke when compared to non-smokers.”

We changed the sentence as suggested by the reviewer.

Line 264: This might be a good place to discuss the results from the sensitivity analyses.

We added discussion about the results of sensitivity analyses.

Line 268: Reference is missing.

Reference was added.

Lines 272–279: Some repetition within the paragraph. Please, truncate.

We truncated the paragraph.

Lines 275 and 277–278: Do you have knowledge about the magnitude and direction of the potential bias?

The magnitude and direction of potential bias could be considered to be shown in sensitivity analyses. Therefore, we added this notion into this sentence.

Lines 278–279: Use same tense within the sentence (now present and imperfect).

We have corrected this grammatical error.

Lines 282–283: Please, do not write in plural (previous studies) if you only have one study as reference. Nonetheless, this is an interesting point.

We corrected this error.

Lines 283–285: The ending comes too late and the phrase is difficult to understand on the first read. Could it be: “By contrast, a study examining young women found that, not the duration of smoking, but the dose of daily smoking…”

We changed the sentence as suggested by the reviewer.

Lines 285–287: Divide into two sentences. Period after “pack years”. A possibility to start the next sentence would be: “Instead, the smoking trajectories…”.

We changed the sentence as suggested by the reviewer.

Line 291: Reference is missing for the part “continuous smoking is related to higher cumulative smoking exposure”.

We changed the sentence as suggested by the reviewer and added a reference.

Lines 293–294: Doesn’t the Table 2 show that, indeed, the highest smoking pack years are among the continuous smokers?

Thank you for this comment. Table 2 indeed shows that. We have changed the text here to highlight this.

Lines 294–295: References are missing concerning the stroke being a rare phenomenon among young adults and concerning the strokes in Finland. Also, delete the preposition “in” in line 295.

References were added and the typo was corrected.

Lines 289–305: This paragraph also discusses a bit about limitations. Could it be placed close to those paragraphs concerning limitations?

We re-arranged the whole discussion part and this paragraph was places differently.

Lines 298–300: Indeed, if most of the participants started smoking at 16 years of age, the number of participants who started smoking at other ages might be that small that it explains why no associations were found. By contrast, I am not sure if the starting age at 16 explains why differences could not be found between the other trajectories and stroke. If trajectory modelling identified the five distinct subgroups, they, in fact, should be distinct – if not by starting age of smoking then for some other reason. Of course, if there are many small trajectory classes, they might not have enough statistical power to show statistical differences. Nonetheless, the 16 years of age might not be the explanation.

We are grateful of this excellent notion. As the reviewer pointed out, the similarities in starting age might not explain the findings. We removed this discussion from the manuscript.

Lines 301–305: Here the discussion between the current and previous findings is interesting. However, in the reference number 32 the researchers have not identified smoking trajectories. They use linear regression in their analyses. Naturally, you may refer to this reference, but use accurate terminology.

We changed the terminology of this sentence. Thank you for noticing this error in our terminology.

Line 307: Delete the first “as”. Also, sometimes you write “pack-years” and sometimes “pack years”. Choose one or the other and use it throughout the text.

We changed the sentence as suggested by the reviewer. Also, we decided to use term “pack-years” throughout the text.

Lines 308–311: If this is the case, why not to identify the smoking trajectories by using the cumulative smoking variable? If not for this study, maybe for the next one to compare which works better.

We would like to thank the reviewer of this comment. This is a very good idea. In future studies, we will compare these two suggested methods of identifying trajectories. This is an interesting research topic and we will continue studying it.

Line 312: “Childhood smoking” was slightly unexpected term since not many children smoke. Please check the reference number 34: do they really have this result concerning childhood smoking or do they mean the duration of smoking from childhood onward?

The reference 34 uses the term “childhood” in their table showing the results of smoking between age 8-17. We changed this term in our manuscript to be more specific age range.

Lines 314–316: Again, do not write in plural (recent studies) if you only have one study as reference. Nonetheless, this is also an interesting point. Thus, what recent studies (line 314)? What previous studies (line 315)? Also, this would be a good place to mention one of the strengths of your study which actually followed the changes in smoking habits from childhood to adulthood. This is a strength of trajectory modelling in longitudinal studies. Other strengths of trajectory modelling can be found, for example, in the following reference:

- Nagin DS. Group-based modeling of development. Cambridge, Mass.: Harvard University Press; 2005.

We have now discussed these suggested strengths of this study. However, to keep the text fluent and consistent, we added this discussion to the section with other strengths. We added references to sentences that were missing them and corrected the plural forms if only one study was cited.

Lines 319–321: Reference is missing.

We have added the references.

Lines 325–328: I ask you to rethink your conclusions. First, indeed, your findings show that cumulative exposure to smoking is a risk factor for stroke. The next part is a bit inconsistent, though: you conclude that the trajectories were not associated with stroke risk even though you have reported in Results and in Discussion that also continuing smoking trajectory is associated with stroke risk. Be consistent with reporting. Also, consider whether you can conclude that starting age of smoking is not crucial for stroke risk if the reason for not finding the association simply was a lack of statistical power. Finally, could you consider adding what is the meaning of these findings for public health or medicine?

Thank you for this comment. We have considered the conclusions and came up with new ones. There were indeed inconsistencies in the previous conclusions. We have also added interpretation of the meaning of findings to the conclusions.

Abbreviations

A list of all the abbreviations used would help the reader, including, for example, NFBC, CRHC, ICH, SAH, TIA, ICD, IS, BMI, etc.

We added a list of abbreviations.

Table 1:

Abbreviations used in tables should always be marked in legends. Usually abbreviations are not used in headings (here ICD). For a reader unfamiliar with ICD, it might not be clear with the first glance that these are codes. Maybe you could add the term “code” under ICD 8, 9 and 10 in the table.

We marked the abbreviations in legends. The abbreviation was removed from the heading. We added the word “code” under ICD 8, 9 and 10.

Table 2:

Period is missing from the end of the heading.

We corrected this typing error.

Number of missing data (n) under “All” and the trajectory classes could be reported as well.

The table is based on imputed data and there are no missing data. We added this notion to the legend of this table.

Terms used for education are not the same as the ones used in the methods.

We corrected them to be same in the text and in table.

Could “kg/m2” and “g/day” be in parentheses?

We added parentheses.

Table 3:

Could “(including IS and TIA)” be marked under “Ischemic stroke” and “(including SAH and ICH)” under “Haemorrhagic stroke”?

We changed these as suggested by the reviewer.

Remember to add in legends the abbreviations used. You should also state in legends with what variables were the models adjusted for and what does the text in bold mean.

Abbreviations and model descriptions were added in legends.

Why the number of strokes before 31 years, between 31-46 and before 46 years are not applicable? Use same words with ages in cumulative smoking (not by and before). 

We apologize the unclear reporting. They are not applicable, because the variables were used as continuous in the model. Above, the numbers of strokes are shown for categorical variables. We have added clarification in legends. Also, the word “before” was corrected to be “by”.

Also, mark the unit which is years.

We have marked them.

It is slightly difficult to understand from the Table 3 how to interpret the HR for cumulative smoking exposure. The text helps to understand it. Would there be a way to make this clearer also in the table?

We added the word “per” in front of smoked pack-years to clarify that the HR is showing the risk per one pack-year. We hope this would clarify the results. Also, we added explanations of analyses in legends.

Figure 1:

Please mark units for x-axis and y-axis. Please mark sample sizes or proportions of participants in each trajectory group. If possible, change the group numbers (1-5) to names of the groups within the figure.

We marked axes and sample sizes and changed the group numbers to group names.

Supplement material 1:

Why there is no information about the % of missing data from variable “use of other drugs at age 14” onward?

These variables were used in multiple imputation model only as predictors. We did not report the amount of missing data for these variables as they were not imputed. 

Apparently 97.2 % had missing data on “age of stroke onset”. Should this variable even be used?

This variable was missing for those people who did not have stroke and for 17 people who had stroke only according to self-report. Therefore, we used this variable in multiple imputation to impute the age of stroke onset for those self-reported stroke diagnoses. For healthy controls, no age of stroke onset was considered.

Supplement material 2: 

Missing all together. Needs to be added.

The information that was supposed to be in Supplement 2 has been added to the main text of the manuscript. Therefore, no Supplement 2 was added.

---

## [Decision Letter · Decision Letter 1]

31 Oct 2019

PONE-D-19-20624R1

Smoking exposure trajectories and risk of stroke until age of 50 years – the Northern Finland Birth Cohort 1966

PLOS ONE

Dear Dr. Rissanen,

Thank you for submitting your manuscript to PLOS ONE. After careful consideration, we feel that it has merit but does not fully meet PLOS ONE’s publication criteria as it currently stands. Therefore, we invite you to submit a revised version of the manuscript that addresses the points raised during the review process.

Based on our reviewer's recommendations, several critical points in your manuscript appear to remain unresolved. Therefore my decision ist still "Major revision".

We would appreciate receiving your revised manuscript by Dec 15 2019 11:59PM. To enhance the reproducibility of your results, we recommend that if applicable you deposit your laboratory protocols in protocols.io, where a protocol can be assigned its own identifier (DOI) such that it can be cited independently in the future. For instructions see: http://journals.plos.org/plosone/s/submission-guidelines#loc-laboratory-protocols

We look forward to receiving your revised manuscript.

Kind regards,

Thomas Behrens

Academic Editor

PLOS ONE

Reviewers' comments:

Reviewer's Responses to Questions

**Comments to the Author**

1. If the authors have adequately addressed your comments raised in a previous round of review and you feel that this manuscript is now acceptable for publication, you may indicate that here to bypass the “Comments to the Author” section, enter your conflict of interest statement in the “Confidential to Editor” section, and submit your "Accept" recommendation.

Reviewer #1: (No Response)

2. Is the manuscript technically sound, and do the data support the conclusions?

Reviewer #1: No

3. Has the statistical analysis been performed appropriately and rigorously? 

Reviewer #1: No

4. Have the authors made all data underlying the findings in their manuscript fully available?

Reviewer #1: No

5. Is the manuscript presented in an intelligible fashion and written in standard English?

Reviewer #1: Yes

6. Review Comments to the Author

Reviewer #1: MAJOR REVISIONS:

I can see the time and effort you have put in improving the manuscript. And indeed, it has improved. The definitions of variables and terms used are clearer. Methods are described better and the reader gets the information needed. The strength of the study still is the long follow-up time and new approach to study the association between smoking and stroke risk.

While the reporting of the trajectory modelling has improved considerably, there is a major limitation that can be seen now that reporting of the modelling is more precise. When interpreting the new Table 2 (Model fit parameters of trajectory models with 1-6 classes) it is difficult to understand why the researchers ended up choosing the 5-class solution as the optimal one for smoking trajectories. It seems that the AIC-values and BIC-values still decrease quite a lot in the 6-class solution. The sample sizes of the classes are still acceptable on the 6th step of the modelling. The proportion of participants in the smallest class is actually higher in the 6-class solution than in the 5-class solution. Additionally, the average posterior probabilities still seem to be very high – which is of course a good thing.

For all of the abovementioned points, I do not find any reasons why the researchers did not continue even further with the modelling and why they chose the 5-class solution over the 6-class solution. This is the reason why I, unfortunately, cannot accept the manuscript to be published as it is. I suggest the manuscript to be rejected if no changes will be done for the trajectory modelling and if the results will not be changed according to the better fitted class solution. This is a reliability issue since the 5-class solution might not give the most reliable description of the smoking behavior of the study population.

Another major issue is the Discussion. While it has improved, there are still some inconsistencies and the discussion and conclusions between your own results and previous literature is not always logical. I have made some comments concerning the Discussion below.

MINOR REVISIONS:

-Terminology: I was actually thinking even on the first round of comments why you use the term smoking exposure, and not simply smoking? I just forgot to mention about it on the previous round. Smoking exposure usually includes active and passive smoking while your study concentrates only on active smoking.

-Typos: There are still typos in the text, for example, on lines 49 and 52 (periods lacking) and on line 58 (word “and” lacking). The typos should be checked once more.

-Lines 63-65: While the use of references has improved, there still are some imprecise ways to refer to other studies. For example, the references 18 and 19 are not studying longitudinal trajectories of smoking which means that the phrase in lines 63-65 is not accurate. Rephrase.

-Line 165: Thank you for adding the descriptions of the variables used. Now the reader knows, for example, that you use measures of leisure-time physical activity (LTPA), not physical activity (PA). I suggest to use the term LTPA instead of PA through-out the text.

-Lines 301-309: The reasoning is hard to follow because the comparing and contrasting is a bit disconnected. According to your results, the smoked-pack years show the dose dependent effect to stroke risk and the continuous smoking trajectory shows the meaning of smoking duration to stroke risk, right? Please be clear what you mean by dose and duration with your own variables and think through the comparing of your results to other studies (e.g., which of your results are in line with previous studies and which are contradictory when considering the effect of dose and duration of smoking).

-Lines 317-320: First, the part “reversal arterial changes” is a bit difficult to understand since you do not say whether you mean positive or negative changes, you just say reversal. The reader will understand what you mean, but you could state it clearer. All in all, this is an interesting sentence. However, I find your sentence following the reference number 43 to be a bit confusing. Your findings actually are contradictory to the findings of the reference 43, right? You found the early quitters to have increased risk for haemorrhagic stroke. Maybe you could think through which of your results to link to the reference number 43.

-Line 326: Would it be better to place your findings concerning the risk of stroke for the continuous smokers after the reference number 44? Just a suggestion.

-Think when to use the term age and when the term time (e.g., lines 303 and 384).

-Reference list: The reference system that you use has not worked as you probably hoped. I saw at least in references 19 and 34 that the names of the authors have not been transferred correctly. It is only fair that the authors’ names are written correctly since this is the way the researchers get acknowledged from their work. Please check the reference list.

-Figure 1: This figure has improved and it is clearer. However, the name “middle quitters” does not really say a lot. Could it be, for example, quitters in mid-age or late quitters? I know it’s tricky to give names to the classes and try an separate them from one another. Still, maybe you could reconsider this name once more.

7. PLOS authors have the option to publish the peer review history of their article (what does this mean?). If published, this will include your full peer review and any attached files.

Reviewer #1: No

---

## [Author Response · Author response to Decision Letter 1]

12 Nov 2019

Dear Academic Editor Thomas Behrens, 

Please find enclosed our revised manuscript entitled “Smoking trajectories and risk of stroke until age of 50 years – the Northern Finland Birth Cohort 1966” (PONE-D-19-20624).

We thank you and the reviewer for the comments and suggestions to improve our manuscript. Below, we present our response to each comment of the reviewer. The original question/request is marked with plain font followed by our response with italic font. We have revised the manuscript according to reviewer’s comments (changes in the manuscript are marked with the ‘track changes’ tool).

We hope that we have satisfactorily responded to the constructive critique and believe those changes have markedly improved our manuscript. We hope that our manuscript could be now accepted for publication.

Yours sincerely,

Ina Rissanen, MD, Ph.D.

Corresponding author

Departments of Neurosurgery and Neurology, Oulu University Hospital,

PL 10, 90029 OYS, Oulu, Finland, and

Julius Center for Health Sciences and Primary Care, University Medical Center Utrecht and Utrecht University,

Huispost nr. STR 6.131, P.O. Box 85500, 3508 GA Utrecht, The Netherlands

Telephone +31618505066

Email ina.rissanen@gmail.com

ORCID 0000-0002-6869-0437 

Smoking trajectories and risk of stroke until age of 50 years – the Northern Finland Birth Cohort 1966 (PONE-D-19-20624)

Reviewer #1: MAJOR REVISIONS:

I can see the time and effort you have put in improving the manuscript. And indeed, it has improved. The definitions of variables and terms used are clearer. Methods are described better and the reader gets the information needed. The strength of the study still is the long follow-up time and new approach to study the association between smoking and stroke risk.

Re: We thank the reviewer for these comments.

While the reporting of the trajectory modelling has improved considerably, there is a major limitation that can be seen now that reporting of the modelling is more precise. When interpreting the new Table 2 (Model fit parameters of trajectory models with 1-6 classes) it is difficult to understand why the researchers ended up choosing the 5-class solution as the optimal one for smoking trajectories. It seems that the AIC-values and BIC-values still decrease quite a lot in the 6-class solution. The sample sizes of the classes are still acceptable on the 6th step of the modelling. The proportion of participants in the smallest class is actually higher in the 6-class solution than in the 5-class solution. Additionally, the average posterior probabilities still seem to be very high – which is of course a good thing.

Re: Previously, we decided to go with the five-class model since we felt that it had more clinical relevance than the six-class model. We hoped the study to be helpful for clinicians and health policy makers when assigning the stroke primary prevention strategies. As suggested by the reviewer, we now present the 6-class solution.

As we now present the results of 6-class trajectory model we also re-considered our previous sensitivity analyses. After consideration we decided not to conduct complete case analyses of non-imputed data as sensitivity analyses as presenting them might be contradictory. Multiple imputation procedure was conducted to prevent bias due to missing data, and therefore, we decided to leave out the sensitivity analyses showing possibly biased results for only those subjects with no missing data. If the editor or the reviewer wishes, we can of course conduct the sensitivity analyses and add them to this manuscript.

For all of the abovementioned points, I do not find any reasons why the researchers did not continue even further with the modelling and why they chose the 5-class solution over the 6-class solution. This is the reason why I, unfortunately, cannot accept the manuscript to be published as it is. I suggest the manuscript to be rejected if no changes will be done for the trajectory modelling and if the results will not be changed according to the better fitted class solution. This is a reliability issue since the 5-class solution might not give the most reliable description of the smoking behavior of the study population.

Re: We now also conducted a seven-class smoking trajectory analysis. Previously we did not go beyond the 6-class solution, because we followed the literature that recommends not going beyond a 6-class solution because they may be more difficult to interpret, less clinically meaningful, and may include one or more classes with small sample size[1]. However, when we modelled a 7-class trajectory, the model did not converge, and we decided to stick with the 6-class solution.

Another major issue is the Discussion. While it has improved, there are still some inconsistencies and the discussion and conclusions between your own results and previous literature is not always logical. I have made some comments concerning the Discussion below.

Re: We made changes to the Discussion section as suggested. We would like to thank you for your comments and suggestions. We have answered the detailed comments below. We truly hope that the Discussion is now more consistent and logical.

MINOR REVISIONS:

1. Terminology: I was actually thinking even on the first round of comments why you use the term smoking exposure, and not simply smoking? I just forgot to mention about it on the previous round. Smoking exposure usually includes active and passive smoking while your study concentrates only on active smoking.

Re: Thank you of pointing this out. We changed the term to be ‘smoking’ instead of ‘smoking exposure’.

2. Typos: There are still typos in the text, for example, on lines 49 and 52 (periods lacking) and on line 58 (word “and” lacking). The typos should be checked once more.

Re: Thank you of noticing these typos. We corrected them.

3. Lines 63-65: While the use of references has improved, there still are some imprecise ways to refer to other studies. For example, the references 18 and 19 are not studying longitudinal trajectories of smoking which means that the phrase in lines 63-65 is not accurate. Rephrase.

Re: We rephrased this sentence to be more precise.

4. Line 165: Thank you for adding the descriptions of the variables used. Now the reader knows, for example, that you use measures of leisure-time physical activity (LTPA), not physical activity (PA). I suggest to use the term LTPA instead of PA through-out the text.

Re: We changed the term ‘physical activity’ into ‘leisure-time physical activity’.

5. Lines 301-309: The reasoning is hard to follow because the comparing and contrasting is a bit disconnected. According to your results, the smoked-pack years show the dose dependent effect to stroke risk and the continuous smoking trajectory shows the meaning of smoking duration to stroke risk, right? Please be clear what you mean by dose and duration with your own variables and think through the comparing of your results to other studies (e.g., which of your results are in line with previous studies and which are contradictory when considering the effect of dose and duration of smoking).

Re: We made changes to this paragraph. We acknowledge that the earlier paragraph was unclear. The smoking trajectory of continuing smoking is supposed to represent the duration of smoking, however, they also have highest number of pack-years that represent dose*duration. The intensity/dose of smoking was not studied separately in this study.

6. Lines 317-320: First, the part “reversal arterial changes” is a bit difficult to understand since you do not say whether you mean positive or negative changes, you just say reversal. The reader will understand what you mean, but you could state it clearer. All in all, this is an interesting sentence. However, I find your sentence following the reference number 43 to be a bit confusing. Your findings actually are contradictory to the findings of the reference 43, right? You found the early quitters to have increased risk for haemorrhagic stroke. Maybe you could think through which of your results to link to the reference number 43.

Re: We changed this paragraph to be more precise. We rephrased the part describing the arterial changes to be more clear for readers. We have also made the following sentences more clear. As the etiologies of ischemic and haemorrhagic strokes are different , our findings are not contradictory to findings of Bogalusa Heart Study. Trajectory classes of those who quitted smoking at young age had increased risk of haemorrhagic stroke that usually is not related to arterial thickness (ICH can be related to arterial thickness, but ususally SAH is not). Also, the onset age of haemorrhagic stroke (especially SAH) is different from that of ischemic strokes.

7. Line 326: Would it be better to place your findings concerning the risk of stroke for the continuous smokers after the reference number 44? Just a suggestion.

Re: We re-arranged the Discussion section as suggested. The findings concerning the stroke risk of continuing smokers come now after the reference 44.

8. Think when to use the term age and when the term time (e.g., lines 303 and 384).

Re: We changed the terms to be more precise.

9. Reference list: The reference system that you use has not worked as you probably hoped. I saw at least in references 19 and 34 that the names of the authors have not been transferred correctly. It is only fair that the authors’ names are written correctly since this is the way the researchers get acknowledged from their work. Please check the reference list.

Re: We corrected the reference list manually as our reference system seemed to fail.

10. Figure 1: This figure has improved and it is clearer. However, the name “middle quitters” does not really say a lot. Could it be, for example, quitters in mid-age or late quitters? I know it’s tricky to give names to the classes and try an separate them from one another. Still, maybe you could reconsider this name once more.

Re: Since we included the 6-class solution we renamed most trajectory classes except for non-smokers and continuing smokers.

References:

1. Nagin DS, Odgers CL. Group-Based Trajectory Modeling in Clinical Research. Annu Rev Clin Psychol. 2010;6: 109-138. doi: 10.1146/annurev.clinpsy.121208.131413.

---

## [Decision Letter · Decision Letter 2]

15 Nov 2019

Smoking trajectories and risk of stroke until age of 50 years 

– the Northern Finland Birth Cohort 1966

PONE-D-19-20624R2

Dear Dr. Rissanen,

We are pleased to inform you that your manuscript has been judged scientifically suitable for publication and will be formally accepted for publication once it complies with all outstanding technical requirements.

With kind regards,

Thomas Behrens

Academic Editor

PLOS ONE

Additional Editor Comments (optional):

Reviewers' comments:

Reviewer's Responses to Questions

**Comments to the Author**

1. If the authors have adequately addressed your comments raised in a previous round of review and you feel that this manuscript is now acceptable for publication, you may indicate that here to bypass the “Comments to the Author” section, enter your conflict of interest statement in the “Confidential to Editor” section, and submit your "Accept" recommendation.

Reviewer #1: All comments have been addressed

2. Is the manuscript technically sound, and do the data support the conclusions?

Reviewer #1: Yes

3. Has the statistical analysis been performed appropriately and rigorously? 

Reviewer #1: Yes

4. Have the authors made all data underlying the findings in their manuscript fully available?

Reviewer #1: Yes

5. Is the manuscript presented in an intelligible fashion and written in standard English?

Reviewer #1: Yes

6. Review Comments to the Author

Reviewer #1: Now it was a pleasure to read your manuscript. The Discussion was clearer and the reasoning for choosing the number of trajectory classes was better. Thank you for your comprehensive answers to my comments. I myself also learned something from them which is always nice in the reviewing process.

7. PLOS authors have the option to publish the peer review history of their article (what does this mean?). If published, this will include your full peer review and any attached files.

Reviewer #1: No

---

## [Editor Report · Acceptance letter]

5 Dec 2019

PONE-D-19-20624R2 

Smoking trajectories and risk of stroke until age of 50 years – the Northern Finland Birth Cohort 1966 

Dear Dr. Rissanen:

I am pleased to inform you that your manuscript has been deemed suitable for publication in PLOS ONE. Congratulations! Your manuscript is now with our production department. 

With kind regards,

on behalf of

Prof. Thomas Behrens 

Academic Editor

PLOS ONE